# Prototypical Contrastive Learning of Unsupervised Representations

**Junnan Li, Pan Zhou, Caiming Xiong, Steven C.H. Hoi**
Salesforce Research
{junnan.li,pzhou,cxiong,shoi}@salesforce.com

## Abstract

This paper presents Prototypical Contrastive Learning (PCL), an unsupervised representation learning method that bridges contrastive learning with clustering. PCL not only learns low-level features for the task of instance discrimination, but more importantly, it encodes semantic structures discovered by clustering into the learned embedding space. Specifically, we introduce prototypes as latent variables to help find the maximum-likelihood estimation of the network parameters in an Expectation-Maximization framework. We iteratively perform E-step as finding the distribution of prototypes via clustering and M-step as optimizing the network via contrastive learning. We propose ProtoNCE loss, a generalized version of the InfoNCE loss for contrastive learning, which encourages representations to be closer to their assigned prototypes. PCL outperforms state-of-the-art instance-wise contrastive learning methods on multiple benchmarks with substantial improvement in low-resource transfer learning. Code and pretrained models are available at https://github.com/salesforce/PCL.

## 1 Introduction

Unsupervised visual representation learning aims to learn image representations from pixels themselves without relying on semantic annotations, and recent advances are largely driven by instance discrimination tasks (Wu et al., 2018; Ye et al., 2019; He et al., 2020; Misra & van der Maaten, 2020; Hjelm et al., 2019; Oord et al., 2018; Tian et al., 2019). These methods usually consist of two key components: image transformation and contrastive loss. Image transformation aims to generate multiple embeddings that represent the same image, by data augmentation (Ye et al., 2019; Bachman et al., 2019; Chen et al., 2020a), patch perturbation (Misra & van der Maaten, 2020), or using momentum features (He et al., 2020). The contrastive loss, in the form of a noise contrastive estimator (Gutmann & Hyvärinen, 2010), aims to bring closer samples from the same instance and separate samples from different instances. Essentially, instance-wise contrastive learning leads to an embedding space where all instances are well-separated, and each instance is locally smooth (*i.e.* input with perturbations have similar representations).

Despite their improved performance, instance discrimination methods share a common weakness: the representation is not encouraged to encode the semantic structure of data. This problem arises because instance-wise contrastive learning treats two samples as a negative pair as long as they are from different instances, regardless of their semantic similarity. This is magnified by the fact that thousands of negative samples are generated to form the contrastive loss, leading to many negative pairs that share similar semantics but are undesirably pushed apart in the embedding space.

In this paper, we propose *prototypical contrastive learning* (PCL), a new framework for unsupervised representation learning that implicitly encodes the semantic structure of data into the embedding space. Figure 1 shows an illustration of PCL. A prototype is defined as "a representative embedding for a group of semantically similar instances". We assign several prototypes of different granularity to each instance, and construct a contrastive loss which enforces the embedding of a sample to be more similar to its corresponding prototypes compared to other prototypes. In practice, we can find prototypes by performing clustering on the embeddings.

We formulate prototypical contrastive learning as an Expectation-Maximization (EM) algorithm, where the goal is to find the parameters of a Deep Neural Network (DNN) that best describes the data

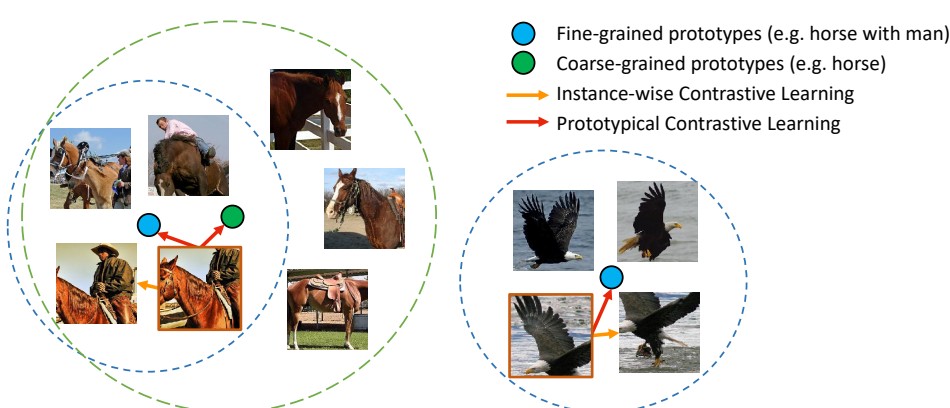

Figure 1: Illustration of Prototypical Contrastive Learning. Each instance is assigned to multiple prototypes with different granularity. PCL learns an embedding space which encodes the semantic structure of data.

distribution, by iteratively approximating and maximizing the log-likelihood function. Specifically, we introduce prototypes as additional latent variables, and estimate their probability in the E-step by performing $k$-means clustering. In the M-step, we update the network parameters by minimizing our proposed contrastive loss, namely *ProtoNCE*. We show that minimizing ProtoNCE is equivalent to maximizing the estimated log-likelihood, under the assumption that the data distribution around each prototype is isotropic Gaussian. Under the EM framework, the widely used instance discrimination task can be explained as a special case of prototypical contrastive learning, where the prototype for each instance is its augmented feature, and the Gaussian distribution around each prototype has the same fixed variance. The contributions of this paper can be summarized as follows:

- We propose prototypical contrastive learning, a novel framework for unsupervised representation learning that bridges contrastive learning and clustering. The learned representation is encouraged to capture the hierarchical semantic structure of the dataset.

- We give a theoretical framework that formulates PCL as an Expectation-Maximization (EM) based algorithm. The iterative steps of clustering and representation learning can be interpreted as approximating and maximizing the log-likelihood function. The previous methods based on instance discrimination form a special case in the proposed EM framework.

- We propose ProtoNCE, a new contrastive loss which improves the widely used InfoNCE by dynamically estimating the concentration for the feature distribution around each prototype. ProtoNCE also includes an InfoNCE term in which the instance embeddings can be interpreted as instance-based prototypes. We provide explanations for PCL from an information theory perspective, by showing that the learned prototypes contain more information about the image classes.

- PCL outperforms instance-wise contrastive learning on multiple benchmarks with substantial improvements in low-resource transfer learning. PCL also leads to better clustering results.

## 2 RELATED WORK

Our work is closely related to two main branches of studies in unsupervised/self-supervised learning: instance-wise contrastive learning and deep unsupervised clustering.

**Instance-wise contrastive learning** (Wu et al., 2018; Ye et al., 2019; He et al., 2020; Misra & van der Maaten, 2020; Zhuang et al., 2019; Hjelm et al., 2019; Oord et al., 2018; Tian et al., 2019; Chen et al., 2020a) aims to learn an embedding space where samples (*e.g.* crops) from the same instance (*e.g.* an image) are pulled closer and samples from different instances are pushed apart. To construct the contrastive loss, positive instance features and negative instance features are generated for each sample. Different contrastive learning methods vary in their strategy to generate instance features. The *memory bank* approach (Wu et al., 2018) stores the features of all samples calculated in the previous step. The *end-to-end* approach (Ye et al., 2019; Tian et al., 2019; Chen et al., 2020a) generates instance features using all samples within the current mini-batch. The *momentum encoder* approach (He et al., 2020) encodes samples on-the-fly by a momentum-updated encoder, and maintains a queue of instance features.

Despite their improved performance, the existing methods based on instance-wise contrastive learning have the following two major limitations, which can be addressed by the proposed PCL framework.

- The task of instance discrimination could be solved by exploiting low-level image differences, thus the learned embeddings do not necessarily capture high-level semantics. This is supported by the fact that the accuracy of instance classification often rapidly rises to a high level (>90% within 10 epochs) and further training gives limited informative signals. A recent study also shows that better performance of instance discrimination could worsen the performance on downstream tasks (Tschannen et al., 2020).
- A sufficiently large number of negative instances need to be sampled, which inevitably yields negative pairs that share similar semantic meaning and should be closer in the embedding space. However, they are undesirably pushed apart by the contrastive loss. Such problem is defined as class collision in (Saunshi et al., 2019) and is shown to hurt representation learning. Essentially, instance discrimination learns an embedding space that only preserves the local smoothness around each instance but largely ignores the global semantic structure of the dataset.

**Deep unsupervised clustering.** Clustering based methods have been proposed for deep unsupervised learning. Xie et al. (2016); Yang et al. (2016); Liao et al. (2016); Yang et al. (2017); Chang et al. (2017); Ji et al. (2019); Gansbeke et al. (2020) jointly learn image embeddings and cluster assignments, but they have not shown the ability to learn transferable representations from a large scale of images. Closer to our work, DeepCluster (Caron et al., 2018) performs iterative clustering and unsupervised representation learning, which is further improved by Zhan et al. (2020) with online clustering. However, our method is conceptually different from DeepCluster. In DeepCluster, the cluster assignments are considered as pseudo-labels and a classification objective is optimized, which results in two weaknesses: (1) the high-dimensional features from the penultimate layer of a ConvNet are not optimal for clustering and need to be PCA-reduced; (2) an additional linear classification layer is frequently re-initialized which interferes with representation learning. In our method, representation learning happens directly in a low-dimensional embedding space, by optimizing a contrastive loss on the prototypes (cluster centroids). Concurrent to our work, SwAV (Caron et al., 2020) also brings together a clustering objective with contrastive learning.

**Self-supervised pretext tasks.** Another line of self-supervised learning methods focus on training DNNs to solve pretext tasks, which usually involve hiding certain information about the input and training the network to recover those missing information. Examples include image inpainting (Pathak et al., 2016), colorization (Zhang et al., 2016; 2017), prediction of patch orderings (Doersch et al., 2015; Noroozi & Favaro, 2016) and image transformations (Dosovitskiy et al., 2014; Gidaris et al., 2018; Caron et al., 2019; Zhang et al., 2019). Compared to heuristic pretext task designs, the proposed PCL is a more general learning framework with better theoretical justification.

## 3 PROTOTYPICAL CONTRASTIVE LEARNING

### 3.1 PRELIMINARIES

Given a training set $X = \{x_1, x_2, ..., x_n\}$ of $n$ images, unsupervised visual representation learning aims to learn an embedding function $f_\theta$ (realized via a DNN) that maps $X$ to $V = \{v_1, v_2, ..., v_n\}$ with $v_i = f_\theta(x_i)$, such that $v_i$ best describes $x_i$. Instance-wise contrastive learning achieves this objective by optimizing a contrastive loss function, such as InfoNCE (Oord et al., 2018; He et al., 2020), defined as:

$$\mathcal{L}_{\text{InfoNCE}} = \sum_{i=1}^{n} - \log \frac{\exp(v_i \cdot v_i'/\tau)}{\sum_{j=0}^{r} \exp(v_i \cdot v_j'/\tau)}, \tag{1}$$

where $v_i$ and $v_i'$ are positive embeddings for instance $i$, and $v_j'$ includes one positive embedding and $r$ negative embeddings for other instances, and $\tau$ is a temperature hyper-parameter. In MoCo (He et al., 2020), these embeddings are obtained by feeding $x_i$ to a momentum encoder parametrized by $\theta'$, $v_i' = f_{\theta'}(x_i)$, where $\theta'$ is a moving average of $\theta$.

In prototypical contrastive learning, we use prototypes $c$ instead of $v'$, and replace the fixed temperature $\tau$ with a per-prototype concentration estimation $\phi$. An overview of our training framework is shown in Figure 2, where clustering and representation learning are performed iteratively at each epoch. Next, we will delineate the theoretical framework of PCL based on EM. A pseudo-code of our algorithm is given in appendix B.

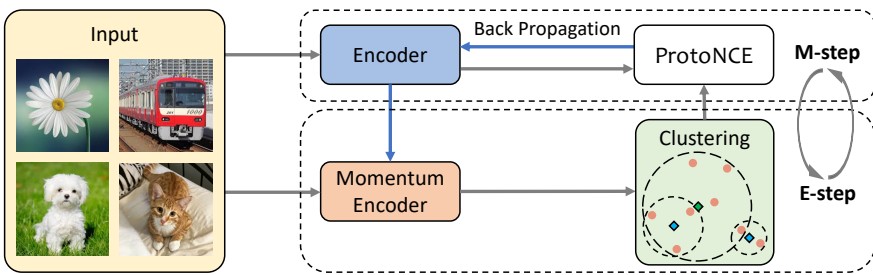

Figure 2: Training framework of Prototypical Contrastive Learning.

## 3.2 PCL AS EXPECTATION-MAXIMIZATION

Our objective is to find the network parameters $\theta$ that maximizes the log-likelihood function of the observed $n$ samples:

$$\theta^* = \arg\max_\theta \sum_{i=1}^n \log p(x_i; \theta) \tag{2}$$

We assume that the observed data $\{x_i\}_{i=1}^n$ are related to latent variable $C = \{c_i\}_{i=1}^k$ which denotes the prototypes of the data. In this way, we can re-write the log-likelihood function as:

$$\theta^* = \arg\max_\theta \sum_{i=1}^n \log p(x_i; \theta) = \arg\max_\theta \sum_{i=1}^n \log \sum_{c_i \in C} p(x_i, c_i; \theta) \tag{3}$$

It is hard to optimize this function directly, so we use a surrogate function to lower-bound it:

$$\sum_{i=1}^n \log \sum_{c_i \in C} p(x_i, c_i; \theta) = \sum_{i=1}^n \log \sum_{c_i \in C} Q(c_i) \frac{p(x_i, c_i; \theta)}{Q(c_i)} \geq \sum_{i=1}^n \sum_{c_i \in C} Q(c_i) \log \frac{p(x_i, c_i; \theta)}{Q(c_i)}, \tag{4}$$

where $Q(c_i)$ denotes some distribution over $c$'s ($\sum_{c_i \in C} Q(c_i) = 1$), and the last step of derivation uses Jensen's inequality. To make the inequality hold with equality, we require $\frac{p(x_i, c_i; \theta)}{Q(c_i)}$ to be a constant. Therefore, we have:

$$Q(c_i) = \frac{p(x_i, c_i; \theta)}{\sum_{c_i \in C} p(x_i, c_i; \theta)} = \frac{p(x_i, c_i; \theta)}{p(x_i; \theta)} = p(c_i; x_i, \theta) \tag{5}$$

By ignoring the constant $-\sum_{i=1}^n \sum_{c_i \in C} Q(c_i) \log Q(c_i)$ in eqn.(4), we should maximize:

$$\sum_{i=1}^n \sum_{c_i \in C} Q(c_i) \log p(x_i, c_i; \theta) \tag{6}$$

**E-step.** In this step, we aim to estimate $p(c_i; x_i, \theta)$. To this end, we perform $k$-means on the features $v_i' = f_{\theta'}(x_i)$ given by the momentum encoder to obtain $k$ clusters. We define prototype $c_i$ as the centroid for the $i$-th cluster. Then, we compute $p(c_i; x_i, \theta) = \mathbb{1}(x_i \in c_i)$, where $\mathbb{1}(x_i \in c_i) = 1$ if $x_i$ belongs to the cluster represented by $c_i$; otherwise $\mathbb{1}(x_i \in c_i) = 0$. Similar to MoCo, we found features from the momentum encoder yield more consistent clusters.

**M-step.** Based on the E-step, we are ready to maximize the lower-bound in eqn.(6).

$$\sum_{i=1}^n \sum_{c_i \in C} Q(c_i) \log p(x_i, c_i; \theta) = \sum_{i=1}^n \sum_{c_i \in C} p(c_i; x_i, \theta) \log p(x_i, c_i; \theta)$$
$$= \sum_{i=1}^n \sum_{c_i \in C} \mathbb{1}(x_i \in c_i) \log p(x_i, c_i; \theta) \tag{7}$$

Under the assumption of a uniform prior over cluster centroids, we have:

$$p(x_i, c_i; \theta) = p(x_i; c_i, \theta)p(c_i; \theta) = \frac{1}{k} \cdot p(x_i; c_i, \theta), \tag{8}$$

where we set the prior probability $p(c_i; \theta)$ for each $c_i$ as $1/k$ since we are not provided any samples. We assume that the distribution around each prototype is an isotropic Gaussian, which leads to:

$$p(x_i; c_i, \theta) = \exp\left(\frac{-(v_i - c_s)^2}{2\sigma_s^2}\right) \Big/ \sum_{j=1}^{k} \exp\left(\frac{-(v_i - c_j)^2}{2\sigma_j^2}\right), \tag{9}$$

where $v_i = f_\theta(x_i)$ and $x_i \in c_s$. If we apply $\ell_2$-normalization to both $v$ and $c$, then $(v-c)^2 = 2-2v \cdot c$. Combining this with eqn.(3, 4, 6, 7, 8, 9), we can write maximum log-likelihood estimation as

$$\theta^* = \arg\min_\theta \sum_{i=1}^{n} -\log\frac{\exp(v_i \cdot c_s/\phi_s)}{\sum_{j=1}^{k} \exp(v_i \cdot c_j/\phi_j)}, \tag{10}$$

where $\phi \propto \sigma^2$ denotes the concentration level of the feature distribution around a prototype and will be introduced later. Note that eqn.(10) has a similar form as the InfoNCE loss in eqn.(1). Therefore, InfoNCE can be interpreted as a special case of the maximum log-likelihood estimation, where the prototype for a feature $v_i$ is the augmented feature $v_i'$ from the same instance (*i.e.* $c = v'$), and the concentration of the feature distribution around each instance is fixed (*i.e.* $\phi = \tau$).

In practice, we take the same approach as NCE and sample $r$ negative prototypes to calculate the normalization term. We also cluster the samples $M$ times with different number of clusters $K = \{k_m\}_{m=1}^{M}$, which enjoys a more robust probability estimation of prototypes that encode the hierarchical structure. Furthermore, we add the InfoNCE loss to retain the property of local smoothness and help bootstrap clustering. Our overall objective, namely **ProtoNCE**, is defined as

$$\mathcal{L}_{\text{ProtoNCE}} = \sum_{i=1}^{n} -\left(\log\frac{\exp(v_i \cdot v_i'/\tau)}{\sum_{j=0}^{r} \exp(v_i \cdot v_j'/\tau)} + \frac{1}{M}\sum_{m=1}^{M}\log\frac{\exp(v_i \cdot c_s^m/\phi_s^m)}{\sum_{j=0}^{r} \exp(v_i \cdot c_j^m/\phi_j^m)}\right). \tag{11}$$

### 3.3 Concentration estimation

The distribution of embeddings around each prototype has different level of concentration. We use $\phi$ to denote the concentration estimation, where a smaller $\phi$ indicates larger concentration. Here we calculate $\phi$ using the momentum features $\{v_z'\}_{z=1}^{Z}$ that are within the same cluster as a prototype $c$. The desired $\phi$ should be small (high concentration) if (1) the average distance between $v_z'$ and $c$ is small, and (2) the cluster contains more feature points (*i.e.* $Z$ is large). Therefore, we define $\phi$ as:

$$\phi = \frac{\sum_{z=1}^{Z}\|v_z' - c\|_2}{Z\log(Z + \alpha)}, \tag{12}$$

where $\alpha$ is a smooth parameter to ensure that small clusters do not have an overly-large $\phi$. We normalize $\phi$ for each set of prototypes $C^m$ such that they have a mean of $\tau$.

In the ProtoNCE loss (eqn.(11)), $\phi_s^m$ acts as a scaling factor on the similarity between an embedding $v_i$ and its prototype $c_s^m$. With the proposed $\phi$, the similarity in a loose cluster (larger $\phi$) are down-scaled, pulling embeddings closer to the prototype. On the contrary, embeddings in a tight cluster (smaller $\phi$) have an up-scaled similarity, thus less encouraged to approach the prototype. Therefore, learning with ProtoNCE yields more balanced clusters with similar concentration, as shown in Figure 3(a). It prevents a trivial solution where most embeddings collapse to a single cluster, a problem that could only be heuristically addressed by data-resampling in DeepCluster (Caron et al., 2018).

### 3.4 Mutual information analysis

It has been shown that minimizing InfoNCE is maximizing a lower bound on the mutual information (MI) between representations $V$ and $V'$ (Oord et al., 2018). Similarly, minimizing the proposed ProtoNCE can be considered as simultaneously maximizing the mutual information between $V$ and all the prototypes $\{V', C^1, ..., C^M\}$. This leads to better representation learning, for two reasons.

First, the encoder would learn the *shared* information among prototypes, and ignore the individual noise that exists in each prototype. The shared information is more likely to capture higher-level semantic knowledge. Second, we show that *compared to instance features, prototypes have a larger mutual information with the class labels*. We estimate the mutual information between the

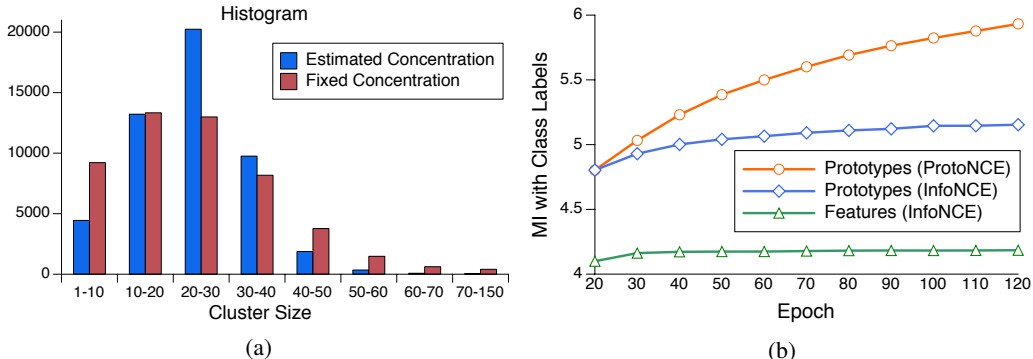

Figure 3: (a) Histogram of cluster size for PCL (#clusters $k$=50000) with fixed or estimated concentration. Using a different $\phi$ for each prototype yields more balanced clusters with similar sizes, which leads to better representation learning. (b) Mutual info between instance features (or their assigned prototypes) and class labels of all images in ImageNet. Compared to InfoNCE, our ProtoNCE learns better prototypes with more semantics.

instance features (or their assigned prototypes) and the ground-truth class labels for all images in ImageNet (Deng et al., 2009) training set, following the method in (Ross, 2014). We compare the obtained MI of our method (ProtoNCE) and that of MoCo (InfoNCE). As shown in Figure 3(b), compared to instance features, the prototypes have a larger MI with the class labels due to the effect of clustering. Furthermore, compared to InfoNCE, training on ProtoNCE can increase the MI of prototypes as training proceeds, indicating that better representations are learned to form more semantically-meaningful clusters.

### 3.5 Prototypes as linear classifier

Another interpretation of PCL can provide more insights into the nature of the learned prototypes. The optimization in eqn.(10) is similar to optimizing the cluster-assignment probability $p(s; x_i, \theta)$ using the cross-entropy loss, where the prototypes $c$ represent weights for a linear classifier. With $k$-means clustering, the linear classifier has a fixed set of weights as the mean vectors for the representations in each cluster, $c = \frac{1}{Z} \sum_{z=1}^{Z} v'_z$. A similar idea has been used for few-shot learning (Snell et al., 2017), where a non-parametric prototypical classifier performs better than a parametric linear classifier.

## 4 Experiments

We evaluate PCL on transfer learning tasks, based on the principle that good representations should transfer with limited supervision and fine-tuning. We follow the settings in MoCo, therefore direct comparisons with MoCo could demonstrate the improvement from the prototypical contrastive loss.

### 4.1 Implementation details

To enable a fair comparison, we follow the same setting as MoCo. We perform training on the ImageNet-1M dataset. A ResNet-50 (He et al., 2016) is adopted as the encoder, whose last fully-connected layer outputs a 128-D and L2-normalized feature. We follow previous works (He et al., 2020; Wu et al., 2018) and perform data augmentation with random crop, random color jittering, random horizontal flip, and random grayscale conversion. We use SGD as our optimizer, with a weight decay of 0.0001, a momentum of 0.9, and a batch size of 256. We train for 200 epochs, where we warm-up the network in the first 20 epochs by only using the InfoNCE loss. The initial learning rate is 0.03, and is multiplied by 0.1 at 120 and 160 epochs. In terms of the hyper-parameters, we set $\tau = 0.1$, $\alpha = 10$, $r = 16000$, and number of clusters $K = \{25000, 50000, 100000\}$. We also experiment with PCL v2 using improvements introduced by Chen et al. (2020a;b), which includes a MLP projection layer, stronger data augmentation with additional Gaussian blur, and temperature $\tau = 0.2$. We adopt faiss (Johnson et al., 2017) for efficient $k$-means clustering. The clustering is performed per-epoch on center-cropped images. We find over-clustering to be beneficial. Recent advances in self-supervised learning have been propelled by huge compute which is inaccessible to many researchers. We instead target a more commonly accessible training resource for PCL with 4 NVIDIA-V100 GPUs and approximately 5 days of training.

| Method | architecture | VOC07 | | | | | Places205 | | | | |
|---|---|---|---|---|---|---|---|---|---|---|---|
| | | $k$=1 | $k$=2 | $k$=4 | $k$=8 | $k$=16 | $k$=1 | $k$=2 | $k$=4 | $k$=8 | $k$=16 |
| Random | ResNet-50 | 8.0 | 8.2 | 8.2 | 8.2 | 8.5 | 0.7 | 0.7 | 0.7 | 0.7 | 0.7 |
| Supervised | | 54.3 | 67.8 | 73.9 | 79.6 | 82.3 | 14.9 | 21.0 | 26.9 | 32.1 | 36.0 |
| Jigsaw | | 26.5 | 31.1 | 40.0 | 46.7 | 51.8 | 4.6 | 6.4 | 9.4 | 12.9 | 17.4 |
| MoCo | ResNet-50 | 31.4 | 42.0 | 49.5 | 60.0 | 65.9 | 8.8 | 13.2 | 18.2 | 23.2 | 28.0 |
| PCL (ours) | | **46.9** | **56.4** | **62.8** | **70.2** | **74.3** | **11.3** | **15.7** | **19.5** | **24.1** | **28.4** |
| SimCLR | | 32.7 | 43.1 | 52.5 | 61.0 | 67.1 | 9.4 | 14.2 | 19.3 | 23.7 | 28.3 |
| MoCo v2 | ResNet-50-MLP | 46.3 | 58.3 | 64.9 | 72.5 | 76.1 | 10.9 | 16.3 | 20.8 | 26.0 | 30.1 |
| PCL v2 (ours) | | **47.9** | **59.6** | **66.2** | **74.5** | **78.3** | **12.5** | **17.5** | **23.2** | **28.1** | **32.3** |

Table 1: **Low-shot image classification** on both VOC07 and Places205 datasets using linear SVMs trained on fixed representations. All methods were pretrained on ImageNet-1M dataset for 200 epochs (except for Jigsaw trained on ImageNet-14M). We vary the number of labeled examples $k$ and report the mAP (for VOC) and accuracy (for Places) across 5 runs. We use the released pretrained model for MoCo, and re-implement SimCLR.

## 4.2 Image classification with limited training data

**Low-shot classification.** We evaluate the learned representation on image classification tasks with few training samples per-category. We follow the setup in Goyal et al. (2019) and train linear SVMs using fixed representations on two datasets: Places205 (Zhou et al., 2014) for scene recognition and PASCAL VOC2007 (Everingham et al., 2010) for object classification. We vary the number $k$ of samples per-class and report the average result across 5 independent runs (standard deviation is reported in appendix C). Table 1 shows the results, in which our method substantially outperforms both MoCo and SimCLR.

**Semi-supervised image classification.** We perform semi-supervised learning experiments to evaluate whether the learned representation can provide a good basis for fine-tuning. Following the setup from Wu et al. (2018); Misra & van der Maaten (2020), we randomly select a subset (1% or 10%) of ImageNet training data (with labels), and fine-tune the self-supervised trained model on these subsets. Table 2 reports the top-5 accuracy on ImageNet validation set. Our method sets a new state-of-the-art under 200 training epochs, outperforming both self-supervised learning methods and semi-supervised learning methods. The standard deviation across 5 runs is low ($< 0.6$ for 1% labels).

| Method | architecture | #pretrain epochs | Top-5 Accuracy | |
|---|---|---|---|---|
| | | | 1% | 10% |
| Random (Wu et al., 2018) | ResNet-50 | - | 22.0 | 59.0 |
| Supervised baseline (Zhai et al., 2019) | ResNet-50 | - | 48.4 | 80.4 |
| *Semi-supervised learning methods:* | | | | |
| Pseudolabels (Zhai et al., 2019) | ResNet-50v2 | - | 51.6 | 82.4 |
| VAT + Entropy Min. (Miyato et al., 2019) | ResNet-50v2 | - | 47.0 | 83.4 |
| $S^4$L Rotation (Zhai et al., 2019) | ResNet-50v2 | - | 53.4 | 83.8 |
| *Self-supervised learning methods:* | | | | |
| Instance Discrimination (Wu et al., 2018) | ResNet-50 | 200 | 39.2 | 77.4 |
| Jigsaw (Noroozi & Favaro, 2016) | ResNet-50 | 90 | 45.3 | 79.3 |
| SimCLR (Chen et al., 2020a) | ResNet-50-MLP | 200 | 56.5 | 82.7 |
| MoCo (He et al., 2020) | ResNet-50 | 200 | 56.9 | 83.0 |
| MoCo v2 (Chen et al., 2020b) | ResNet-50-MLP | 200 | 66.3 | 84.4 |
| PCL v2 (ours) | ResNet-50-MLP | 200 | 73.9 | 85.0 |
| PCL (ours) | ResNet-50 | 200 | **75.3** | **85.6** |
| PIRL (Misra & van der Maaten, 2020) | ResNet-50 | 800 | 57.2 | 83.8 |
| SimCLR Chen et al. (2020a) | ResNet-50-MLP | 1000 | 75.5[†] | 87.8[†] |
| BYOL (Grill et al., 2020) | ResNet-50-MLP$_{\text{big}}$ | 1000 | 78.4[†] | 89.0[†] |
| SwAV (Caron et al., 2020) | ResNet-50-MLP | 800 | 78.5[‡] | 89.9[‡] |

Table 2: **Semi-supervised learning** on ImageNet. We report top-5 accuracy on the ImageNet validation set of self-supervised models that are finetuned on 1% or 10% of labeled data. [‡]: SimCLR, BYOL, and SwAV use a large batch size of 4096. [‡]: SwAV uses multi-crop augmentation.

### 4.3 IMAGE CLASSIFICATION BENCHMARKS

**Linear classifiers.** Next, we train linear classifiers on fixed image representations using the entire labeled training data. We evaluate the performance of such linear classifiers on three datasets: ImageNet, VOC07, and Places205. Table 3 reports the results. PCL outperforms MoCo under direct comparison, which demonstrate the advantage of the proposed prototypical contrastive loss.

| Method | architecture (#params) | #pretrain epochs | Dataset | | |
|---|---|---|---|---|---|
| | | | ImageNet | VOC07 | Places205 |
| Jigsaw (Noroozi & Favaro, 2016) | R50 (24M) | 90 | 45.7 | 64.5 | 41.2 |
| Rotation (Gidaris et al., 2018) | R50 (24M) | – | 48.9 | 63.9 | 41.4 |
| DeepCluster (Caron et al., 2018) | VGG(15M) | 100 | 48.4 | 71.9 | 37.9 |
| BigBiGAN (Donahue & Simonyan, 2019) | R50 (24M) | – | 56.6 | – | – |
| InstDisc (Wu et al., 2018) | R50 (24M) | 200 | 54.0 | – | 45.5 |
| MoCo (He et al., 2020) | R50 (24M) | 200 | 60.6 | 79.2* | 48.9* |
| PCL (ours) | R50 (24M) | 200 | **61.5** | **82.3** | **49.2** |
| SimCLR (Chen et al., 2020a) | R50-MLP (28M) | 200 | 61.9 | – | – |
| MoCo v2 (Chen et al., 2020b) | R50-MLP (28M) | 200 | 67.5 | 84.0* | 50.1* |
| PCL v2 (ours) | R50-MLP (28M) | 200 | **67.6** | **85.4** | **50.3** |
| LocalAgg (Zhuang et al., 2019) | R50 (24M) | 200 | $60.2^\dagger$ | – | $50.1^\dagger$ |
| SelfLabel (Asano et al., 2020) | R50 (24M) | 400 | 61.5 | – | – |
| CPC (Oord et al., 2018) | R101 (28M) | – | 48.7 | – | – |
| CMC (Tian et al., 2019) | $R50_{L+ab}$ (47M) | 280 | 64.0 | – | – |
| PIRL (Misra & van der Maaten, 2020) | R50 (24M) | 800 | 63.6 | 81.1 | 49.8 |
| AMDIM (Bachman et al., 2019) | Custom (626M) | 150 | $68.1^\dagger$ | – | $55.0^\dagger$ |
| SimCLR (Chen et al., 2020a) | R50-MLP (28M) | 1000 | $69.3^\dagger$ | $80.5^\dagger$ | – |
| BYOL (Grill et al., 2020) | R50-MLP$_{big}$(35M) | 1000 | $74.3^\dagger$ | - | – |
| SwAV (Caron et al., 2020) | R50-MLP (28M) | 800 | $75.3^\dagger$ | $88.9^\dagger$ | $56.7^\dagger$ |

Table 3: **Image classification with linear models.** We report top-1 accuracy. Numbers with * are from released pretrained model; all other numbers are adopted from corresponding papers.
$\dagger$: LocalAgg uses 10-crop evaluation. ADMIM uses FastAutoAugment (Lim et al., 2019) that is supervised by ImageNet labels. SwAV uses multi-crop augmentation. SimCLR, BYOL, and SwAV use a large batch size of 4096.

**KNN classifiers.** We perform k-nearest neighbor (kNN) classification on ImageNet. For a query image with feature $v$, we take its top $k$ nearest neighbors from the momentum features, and perform weighted-combination of their labels where the weights are calculated by $\exp(v \cdot v_i'/\tau)$. Table 4 reports the accuracy. Our method substantially outperforms previous methods.

| Method | Inst. Disc. (Wu et al., 2018) | MoCo (He et al., 2020) | LA (Zhuang et al., 2019) | PCL (ours) |
|---|---|---|---|---|
| Accuracy | 46.5 | 47.1 | 49.4 | **54.5** |

Table 4: **Image classification with kNN classifiers** using ResNet-50 features on ImageNet.

### 4.4 CLUSTERING EVALUATION

In Table 5, we evaluate the $k$-means clustering performance on ImageNet using representations learned by different methods. PCL leads to substantially higher adjusted mutual information (AMI) score. Details are given in appendix F.

| Method | DeepCluster (Caron et al., 2018) | MoCo (He et al., 2020) | PCL (ours) |
|---|---|---|---|
| AMI | 0.281 | 0.285 | **0.410** |

Table 5: AMI score for k-means clustering ($k = 25000$) on ImageNet representation.

### 4.5 OBJECT DETECTION

We assess the representation on object detection. Following Goyal et al. (2019), we train a Faster R-CNN (Ren et al., 2015) on VOC07 or VOC07+12, and evaluate on the test set of VOC07. We keep the pretrained backbone frozen to better evaluate the learned representation, and use the same

schedule for all methods. Table 6 reports the average mAP across three runs. Our method substantially closes the gap between self-supervised methods and supervised training. In appendix D, we show the results for fine-tuning the pretrained model for object detection and instance segmentation on COCO (Lin et al., 2014), where PCL outperforms both MoCo and supervised training.

| Method | Pretrain Dataset | Architecture | Training data | |
| --- | --- | --- | --- | --- |
| | | | VOC07 | VOC07+12 |
| Supervised | ImageNet-1M | Resnet-50-FPN | 72.8 | 79.3 |
| MoCo (He et al., 2020) | ImageNet-1M | Resnet-50-FPN | 66.4 | 73.5 |
| PCL (ours) | ImageNet-1M | Resnet-50-FPN | **71.7** | **78.5** |

Table 6: **Object detection** for frozen `conv` body on VOC using Faster R-CNN.

## 5 VISUALIZATION OF LEARNED REPRESENTATION

In Figure 4, we visualize the unsupervised learned representation of ImageNet training images using t-SNE (Maaten & Hinton, 2008). Compared to the representation learned by MoCo, the representation learned by the proposed PCL forms more separated clusters, which also suggests representation of lower entropy.

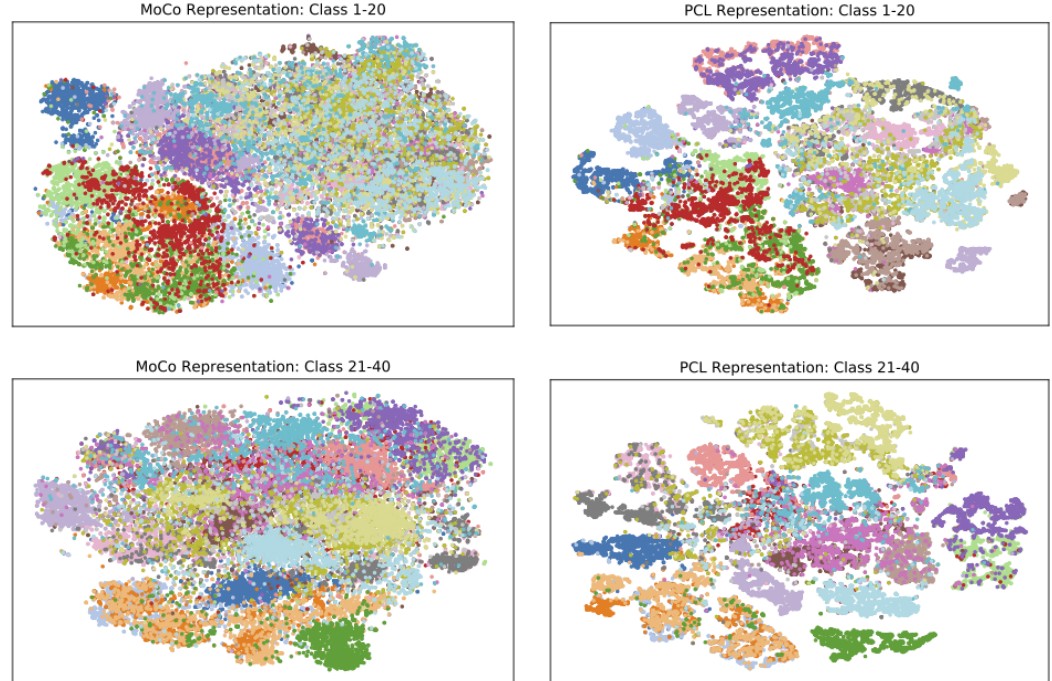

Figure 4: T-SNE visualization of the unsupervised learned representation for ImageNet training images from the first 40 classes. Left: MoCo; Right: PCL (ours). Colors represent classes.

## 6 CONCLUSION

This paper proposes Prototypical Contrastive Learning, a generic unsupervised representation learning framework that finds network parameters to maximize the log-likelihood of the observed data. We introduce prototypes as latent variables, and perform iterative clustering and representation learning in an EM-based framework. PCL learns an embedding space which encodes the semantic structure of data, by training on the proposed ProtoNCE loss. Our extensive experiments on multiple benchmarks demonstrate the advantage of PCL for unsupervised representation learning.

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

## APPENDIX A    ABLATION ON PROTONCE

The proposed loss in eqn.(11) contains two terms: the instance-wise contrastive loss and the proposed prototypical contrastive loss. Here we study the effect of each term on representation learning. Table 7 reports the results for low-resource fine-tuning and linear classification on ImageNet. The prototypical term plays an important role, especially in the fine-tuning experiment. The warm-up also improves the result by bootstrapping the clustering with better representations.

| Method | 1% fine-tuning (top-5 acc.) | linear classification (top-1 acc.) |
|---|---|---|
| instance only | 56.9 | 60.6 |
| proto only (w/o warm-up) | 60.7 | 60.4 |
| proto only (w/ warm-up) | 72.3 | 60.9 |
| instance + proto (w/o warm-up) | 74.6 | 61.3 |
| instance + proto (w/ warm-up) | **75.3** | **61.5** |

Table 7: Effect of instance-wise contrastive loss and prototypical contrastive loss.

## APPENDIX B    PSEUDO-CODE FOR PROTOTYPICAL CONTRASTIVE LEARNING

---
**Algorithm 1:** Prototypical Contrastive Learning.

---

1  **Input:** encoder $f_\theta$, training dataset $X$, number of clusters $K = \{k_m\}_{m=1}^{M}$
2  $\theta' = \theta$                                  // initialize momentum encoder as the encoder
3  **while** not MaxEpoch **do**
        /* E-step */
4      $V' = f_{\theta'}(X)$                    // get momentum features for all training data
5      **for** $m = 1$ **to** $M$ **do**
6          $C^m = k-\text{means}(V', k_m)$                // cluster $V'$ into $k_m$ clusters, return prototypes
7          $\phi_m = \text{Concentration}(C^m, V')$  // estimate the distribution concentration around each prototype with Equation 12
8      **end**
        /* M-step */
9      **for** $x$ in $\text{Dataloader}(X)$ **do**                          // load a minibatch $x$
10         $v = f_\theta(x), v' = f_{\theta'}(x)$      // forward pass through encoder and momentum encoder
11         $\mathcal{L}_{\text{ProtoNCE}}(v, v', \{C^m\}_{m=1}^{M}, \{\phi_m\}_{m=1}^{M})$    // calculate loss with Equation 11
12         $\theta = \text{SGD}(\mathcal{L}_{\text{ProtoNCE}}, \theta)$                          // update encoder parameters
13         $\theta' = 0.999 * \theta' + 0.001 * \theta$                          // update momentum encoder
14     **end**
15 **end**

---

## APPENDIX C    STANDARD DEVIATION FOR LOW-SHOT CLASSIFICATION

In Table 7, we report the standard deviation for the low-shot classification experiments in Table 1.

| Method | VOC07 | | | | | Places205 | | | | |
|---|---|---|---|---|---|---|---|---|---|---|
| | $k$=1 | $k$=2 | $k$=4 | $k$=8 | $k$=16 | $k$=1 | $k$=2 | $k$=4 | $k$=8 | $k$=16 |
| PCL | 4.06 | 2.65 | 2.21 | 0.49 | 0.39 | 0.24 | 0.23 | 0.13 | 0.07 | 0.05 |
| PCL v2 | 4.12 | 2.70 | 2.17 | 0.54 | 0.38 | 0.26 | 0.23 | 0.12 | 0.08 | 0.04 |

Table 8: Standard deviation across 5 runs for low-shot image classification experiments.

## APPENDIX D  COCO OBJECT DETECTION AND SEGMENTATION

Following the experiment setting in (He et al., 2020), we use Mask R-CNN (He et al., 2017) with C4 backbone. We finetune all layers end-to-end on the COCO train2017 set and evaluate on val2017. The schedule is the default $2\times$ in (Girshick et al., 2018). PCL outperforms both MoCo (He et al., 2020) and supervised pre-training in all metrics.

| Method | $AP^{bb}$ | $AP^{bb}_{50}$ | $AP^{bb}_{75}$ | $AP^{mk}$ | $AP^{mk}_{50}$ | $AP^{mk}_{75}$ |
|---|---|---|---|---|---|---|
| Supervised | 40.0 | 59.9 | 43.1 | 34.7 | 56.5 | 36.9 |
| MoCo (He et al., 2020) | 40.7 | 60.5 | 44.1 | 35.4 | 57.3 | 37.6 |
| PCL (ours) | **41.0** | **60.8** | **44.2** | **35.6** | **57.4** | **37.8** |

Table 9: Object detection and instance segmentation fine-tuned on COCO. We evaluate bounding-box AP ($AP^{bb}$) and mask AP ($AP^{mk}$) on val2017.

## APPENDIX E  TRAINING DETAILS FOR TRANSFER LEARNING EXPERIMENTS

For training linear SVMs on Places and VOC, we follow the procedure in (Goyal et al., 2019) and use the LIBLINEAR (Fan et al., 2008) package. We preprocess all images by resizing to 256 pixels along the shorter side and taking a $224 \times 224$ center crop. The linear SVMs are trained on the global average pooling features of ResNet-50.

For image classification with linear models, we use the pretrained representations from the global average pooling features (2048-D) for ImageNet and VOC, and the conv5 features (averaged pooled to $\sim$9000-D) for Places. We train a linear SVM for VOC, and a logistic regression classifier (a fully-connected layer followed by softmax) for ImageNet and Places. The logistic regression classifier is trained using SGD with a momentum of 0.9. For ImageNet, we train for 100 epochs with an initial learning rate of 10 and a weight decay of 0. Similar hyper-parameters are used by (He et al., 2020). For Places, we train for 40 epochs with an initial learning rate of 0.3 and a weight decay of 0.

For semi-supervised learning, we finetune ResNet-50 with pretrained weights on a subset of ImageNet with labels. We optimize the model with SGD, using a batch size of 256, a momentum of 0.9, and a weight decay of 0.0005. We apply different learning rate to the ConvNet and the linear classifier. The learning rate for the ConvNet is 0.01, and the learning rate for the classifier is 0.1 (for 10% labels) or 1 (for 1% labels). We train for 20 epochs, and drop the learning rate by 0.2 at 12 and 16 epochs.

For object detection on VOC, We use the R50-FPN backbone for the Faster R-CNN detector available in the `MMdetection` (Chen et al., 2019) codebase. We freeze all the `conv` layers and also fix the BatchNorm parameters. The model is optimized with SGD, using a batch size of 8, a momentum of 0.9, and a weight decay of 0.0001. The initial learning rate is set as 0.05. We finetune the models for 15 epochs, and drop the learning rate by 0.1 at 12 epochs.

## APPENDIX F  EVALUATION OF CLUSTERING

In order to evaluate the quality of the clusters produced by PCL, we compute the adjusted mutual information score (AMI) (Nguyen et al., 2010) between the clusterings and the ground-truth labels for ImageNet training data. AMI is adjusted for chance which accounts for the bias in MI to give high values to clusterings with a larger number of clusters. AMI has a value of 1 when two partitions are identical, and an expected value of 0 for random (independent) partitions. In Figure 5, we show the AMI scores for three clusterings obtained by PCL, with number of clusters $K = \{25000, 50000, 100000\}$. In Table 5, we show that compared to DeepCluster (Caron et al., 2018) and MoCo (He et al., 2020), PCL produces clusters of substantially higher quality.

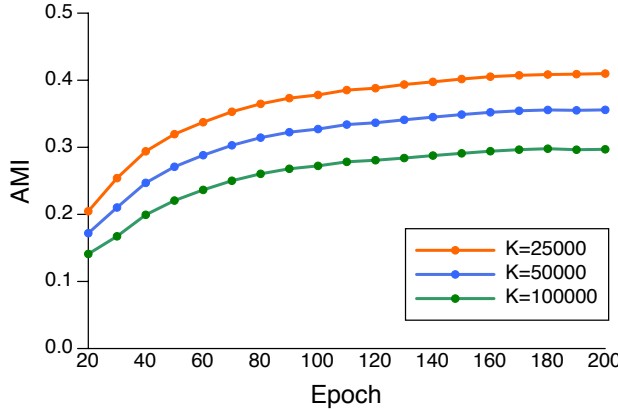

Figure 5: Adjusted mutual information score between the clusterings generated by PCL and the ground-truth labels for ImageNet training data.

## APPENDIX G    CONVERGENCE PROOF

Here we provide the proof that the proposed PCL would converge. Suppose let

$$F(\theta) = \sum_{i=1}^{n} \log p(x_i; \theta) = \sum_{i=1}^{n} \log \sum_{c_i \in C} p(x_i, c_i; \theta) = \sum_{i=1}^{n} \log \sum_{c_i \in C} Q(c_i) \frac{p(x_i, c_i; \theta)}{Q(c_i)} \\ \geq \sum_{i=1}^{n} \sum_{c_i \in C} Q(c_i) \log \frac{p(x_i, c_i; \theta)}{Q(c_i)}. \tag{13}$$

We have shown in Section 3.2 that the above inequality holds with equality when $Q(c_i) = p(c_i; x_i, \theta)$.

At the $t$-th E-step, we have estimated $Q^t(c_i) = p(c_i; x_i, \theta^t)$. Therefore we have:

$$F(\theta^t) = \sum_{i=1}^{n} \sum_{c_i \in C} Q^t(c_i) \log \frac{p(x_i, c_i; \theta^t)}{Q^t(c_i)}. \tag{14}$$

At the $t$-th M-step, we fix $Q^t(c_i) = p(c_i; x_i, \theta^t)$ and train parameter $\theta$ to maximize Equation 14. Therefore we always have:

$$F(\theta^{t+1}) \geq \sum_{i=1}^{n} \sum_{c_i \in C} Q^t(c_i) \log \frac{p(x_i, c_i; \theta^{t+1})}{Q^t(c_i)} \geq \sum_{i=1}^{n} \sum_{c_i \in C} Q^t(c_i) \log \frac{p(x_i, c_i; \theta^t)}{Q^t(c_i)} = F(\theta^t). \tag{15}$$

The above result suggests that $F(\theta^t)$ monotonously increase along with more iterations. Hence the algorithm will converge.

## APPENDIX H    VISUALIZATION OF CLUSTERS

In Figure 6, we show ImageNet training images that are randomly chosen from clusters generated by the proposed PCL. PCL not only clusters images from the same class together, but also finds fine-grained patterns that distinguish sub-classes, demonstrating its capability to learn useful semantic representations.

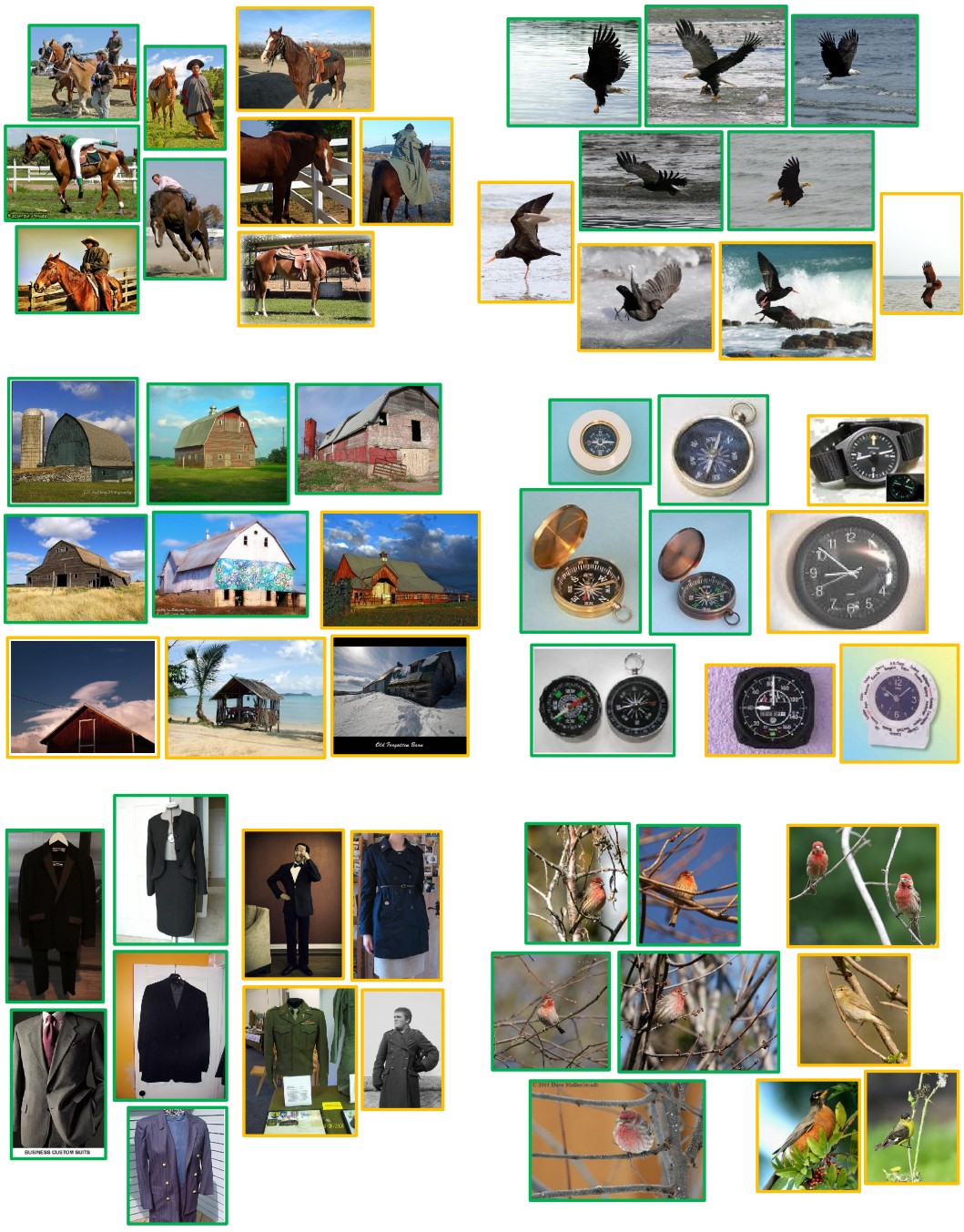

Figure 6: Visualization of randomly chosen clusters generated by PCL. Green boarder marks top-5 images that are closest to fine-grained prototypes ($K = 100k$). Orange boarder marks images randomly chosen from coarse-grained clusters ($K = 50k$) that also cover the same green images. PCL can discover hierarchical semantic structures within the data (*e.g.* images with horse and man form a fine-grained cluster within the coarse-grained horse cluster.)

