# OpenReview forum: "Prototypical Contrastive Learning of Unsupervised Representations"
_ICLR.cc/2021/Conference — ICLR 2021 Poster_

### Official Review · AnonReviewer2 · 2020-10-22
**Review for "Prototypical Contrastive Learning of Unsupervised Representations"**

**Rating:** 7
**Confidence:** 4

**Review:**

### Summary
In this paper, the authors propose a method for contrastive learning inspired on current instance discrimination approaches. The propose method, called Prototypical Contrastive Learning, can be seen as an augmentation of MoCo where, besides discriminating each instance, they also introduce prototypes as latent variables. A EM-like algorithm is introduce to train the model with latent variables.
The model is trained by considering both instance discrimination (as in MoCo) and prototypical discrimination. The authors provide results in different setting (low-shot image classification, semi-supervised learning and image classification with fixed features).

### Strengths
+ The paper is nicely written and provides a nice read.
+ The paper is well motivated and points to valid issue with current issues with (unsupervised) contrastive learning approaches. The idea of providing latent 'prototype' variables to capture the semantic structure of data.
+ The model is built on top of MoCo and the experiments show constant improvement over MoCo on different range of experiments.

### Weakness
- My main issue with this paper is the lack of comparison with current similar model, SwAV (Caron et al., NeurIPS20). Although quickly mentioned on the related works section, the authors do compare with them on experimental section.
- I feel that some ablation studies to better understand the components of the method is missing (the number of clusters, the importance of the concentration estimation and the choices of its parameters, etc.). The qualitative visualization from the appendix are very nice and it is unfortunate that they cannot go on main manuscript.

### Comments
Although I generally like the paper, i dont understand why the authors did not include most recent methods in the experimental results (eg. BYOL, SwAV, etc). In particular, no mention or comparison with SwAV---a method that is similar to the proposed approach---appears in experimental section.
I don't particularly mind the fact that SwAV outperform the proposed approach (the fact the method is built on the top of MoCo AND outperforms it is not bad). However, I dont think the method should be omitted from the experimental results. I would be much more happy to see more comparison (similarities/differences) between the two approaches.

Because of this, my current rate is "Marginally below acceptance threshold"

### Post-rebuttal updates
Given the answer of the authors. I will raise my reviews to 7 assuming the authors will add more about comapring with current sota methods, as discussed on this review.

---

> ### Author Response · Authors · 2020-11-14
> **Response to Reviewer #2**
>
> We thank the reviewer for the insightful and valuable comments.  Here are our responses to the reviewers concerns.
> - Following the valuable suggestion, we have included results from BYOL and SwAV in Table 2 and Table 3. Note that both BYOL and SwAV use a huge computation resource with longer training epochs, larger batch size, and multi-crop augmentations (only SwAV). Moreover, the Arxiv versions of BYOL and SwAV were released after we released PCL online.
> From a technical perspective, SwAV introduces an optimization objective per-minibatch which performs online clustering on a subset of training samples. Different from SwAV, PCL performs offline clustering on all samples. Hence, PCL would produce more semantically meaningful clusters that capture the global pattern, whereas SwAV would produce more flexible clusters that capture local patterns. Furthermore, PCL introduces the cluster-varying concentration estimation which replaces the fixed temperature in SwAV.
> - The concentration estimation is very important. With a fixed concentration, the representations would be skewed due to cluster imbalance (as shown in Figure 3a), and the performance would significantly decrease. Due to limitations in computation budget, we have not performed an extensive search for the hyper-parameters including number of clusters, parameters for concentration estimation, etc. We have provided the ablation study on the proposed loss in Table 7, and will try our best to conduct more ablation studies. We appreciate the reviewer’s approval of our visualization, and have moved the t-SNE visualization to the main text due to the allowance of an additional page.

---

### Official Review · AnonReviewer3 · 2020-10-25
**A good exploration with a solid set of experiments. The major concern is significance as the field advances fast**

**Rating:** 6
**Confidence:** 5

**Review:**

The paper mainly injects clustering into the instance discrimination work, MoCo, for unsupervised representation learning. The main idea is to maintain a list of cluster centers (i.e., prototypes), and use them to estimate the similarity between a key and query. Because the cluster centers also need to be learned, the proposed method is an iterative, EM-like algorithm where the clusters are produced every now and then.

From the arXiv upload time, this is an improved version of the manuscript submitted for NeurIPS 2020. Back then I was pretty interested in this work and was able to reproduce some of the improvements (e.g., some minor improvements on ImageNet linear evaluation) observed in the paper using my own implementation. So this paper did a pretty good exploration and I would trust the results in the paper. However, the main concern is on significance now, as there are concurrent submissions like BYOL (which claims to implicitly perform bootstrapping representation learning), or more closely SwAV (an online-clustering algorithm that also aims at unsupervised learning). Both BYOL and SwAV are accepted at NeurIPS already with better performance on several evaluation benchmarks (e.g. semi-supervised setting, BYOL reports 78.4 top-5 accuracy with 1% labels), yet unfortunately this work does not. Nevertheless, I would still give an acceptance rating as:

1) This is solid work, I have personally implemented this work and the observation is quite consistent with what's described in the paper, at least for some major experiments.
2) Doing an EM-style clustering with momentum encoder/key is indeed something that has never been explored to my knowledge, and there is some novelty to it.
3) It also proposes a prototype-specific T, which is quite new and well-motivated.

For writing and organization: I think the paper is written quite clearly (as I was able to write code based on this), and since this is already an improved version for the NeurIPS submission, I believe the clarity/organization has improved more.

Suggestions/questions:
1) Please include more recent numbers from other papers like BYOL or SwAV, for completeness of the assessment of the value of the work.
2) Does more recent method (BYOL/SwAV) also have "clustering" effect? Like the mutual information with class labels on ImageNet?
3) How  much does the "cluster-varying T" help performance? Can a single, global T used for protoNCE?

---

> ### Author Response · Authors · 2020-11-14
> **Response to Reviewer #3**
>
> We thank the reviewer for the insightful and positive comments. Here are our responses to the reviewers comments.
>
> 1. Following the suggestion, we have included results from BYOL and SwAV in Table 2 and Table 3.
> 2. SwAV introduces an online optimization which performs clustering on a subset of training samples. We perform k-means clustering on top of the representations, using the 200-epoch model released by the authors. The adjusted mutual information (AMI) score is 40.2, similar to 41.0 as produced by PCL. Hence we hypothesize that SwAV has a similar clustering effect. In terms of BYOL, since it does not optimize for a clustering objective, we hypothesize that the representations learned by BYOL will be more evenly distributed similar to MoCo.
> 3. The cluster-varying temperature is crucial for good performance due to its ability to address the imbalance in cluster sizes. Using a global temperature results in more imbalanced clusters (as shown in Figure 3a) and a skewed distribution of representations. The final performance also drops by a considerable amount if a global temperature is used. The imbalance issue could be partially mitigated by a heuristic sampling strategy based on cluster size, but cluster-varying temperature is a more principled solution.

---

### Official Review · AnonReviewer1 · 2020-10-27
**Interesting approach, empirical results not persuasive enough**

**Rating:** 5
**Confidence:** 4

**Review:**

##########################################################################

Summary:

The paper proposes a new self-supervised representation learning loss which is a combination of InfoNCE and EM-clustering. Rather than encouraging similarity to another augmentation of the same image, it clusters images and encourages similarity of images to their cluster centers. The evaluated approach ProtoNCE actually consists of a sum of the standard InfoNCE loss and the proposed novel loss. The approach is evaluated on the most common ImageNet linear classification benchmark and a few others: ImageNet semi-supervised, VOC07, Places 205, including some detection tasks. The paper compares to reasonable baselines like MoCo-v2 and SimCLR.


##########################################################################

Reasons for score:

The proposed approach seems relatively complicated to implement compared to baseline MoCo-v2 while not being sufficiently better on the most established ImageNet linear eval benchmark. MoCo-v2 produces 67.5±0.1 (five runs from github repo: {67.7, 67.6, 67.4, 67.6, 67.3}, see https://github.com/facebookresearch/moco) while the proposed method gets 67.6 - not a very persuasive advantage. Moreover, the approach essentially compares with with 200-epoch ablation of MoCo-v2 without comparing to the main 800-epoch run. I couldn't find other numbers in MoCo-v2 paper https://arxiv.org/pdf/2003.04297.pdf surpassed by the numbers in this paper - although both papers evaluate on detection as well.


##########################################################################

Pros:

1. Interesting idea to compare images to EM cluster centers.
2. Reasonable amount of experiments.

##########################################################################

Cons:

1. Results are within variance of MoCo-v2 number on ImageNet linear eval. Other numbers from MoCo-v2 paper are not directly compared against.
2. It does not follow from the title/abstract/introduction that the evaluated approach is actually a sum of the standard InfoNCE and the novel loss. This is a clarity issue which needs to be addressed.
3. The paper seems to make an implicit assumption about the computational budget under which it tries to obtain the best possible results. For example, it doesn't try to beat 1000-epoch-big-batch results of SimCLR. I don't have an issue with this assumption, but it needs to be directly articulated early on. Something like "recent advances in self-supervised learning were propelled by huge compute, which is not available to the majority of labs; we instead target more commonly reproducible/achievable training regimes - X GPUs, Y days". For example, MoCo-v2 paper makes these compute considerations clear upfront, as early as abstract - which is good for the reader.

##########################################################################

Questions during rebuttal period:

Please address and clarify the cons above.

#########################################################################

Minor suggestions and typos:

(1) c_s is not explained anywhere and it's a quite confusing notation for the center of the cluster which x_i belongs to

(2) the fact that the paper uses data augmentations is not mentioned in the main text at all

(3) "where v'i is a positive embedding for instance i," - should be without '?

(4) "PCL not only learns low-level features for the task of instance discrimination, but
more importantly, it implicitly encodes semantic structures of the data into the
learned embedding space." - it is quite confusing to talk about semantic structures here, without clarifying what is meant by that. The clusters might have good mutual information with respect to unseen labels, but there is still no way to know that dog is a dog until an example of a dog is seen.

(5) it might be nice (although not required by ICLR rules) to mention BYOL https://arxiv.org/abs/2006.07733 in something like Concurrent Work section

---

> ### Author Response · Authors · 2020-11-14
> **Response to Reviewer #1**
>
> We thank the reviewer for the insightful and valuable comments. Here are our responses to the reviewer's concerns.
>
> 1. Implementation complexity.
> \
> Our method is straightforward to implement, as suggested by Reviewer 3 who has re-implemented our method and achieved similar improvements.
> 2. Comparison to MoCo-v2.
> \
> All of our experimental results on MoCo-v2 that are not reported in their original papers are produced using the officially released pre-trained models. The comparisons are done in a fair way with the same evaluation script as PCL. We would like to highlight that compared to the linear evaluation setting with abundant labels, PCL and PCL-v2 demonstrate much stronger advantages on low-resource transfer learning, as shown in the few-shot classification results and semi-supervised learning results in Table 1 and Table 2. We believe that low-resource transfer learning has high practical values for self-supervised representation learning and is commonly used in real applications.
> 3. Clarification of the proposed loss.
> \
> The InfoNCE loss can be interpreted as a special case of our ProtoNCE loss, where each instance is a prototype, and the feature distribution around each instance has the same variance. Following the reviewer’s suggestion, we have improved the paper clarity by adding the following sentence in introduction: “ProtoNCE also includes an InfoNCE term in which the instance embeddings can be interpreted as instance-based prototypes.”, and will further revise it.
> 4. Computation budget.
> \
> We appreciate the reviewer’s understanding of the computation budget. We have updated the paper to specify this practical setting in Section 4.1.
> 5. We appreciate the minor suggestions and have improved the paper accordingly. With the allowance of an additional page, we have included the data augmentation details in the main text. BYOL has also been added.

---

### Official Review · AnonReviewer4 · 2020-10-31
**A good paper presenting prototypical contrastive learning that combines deep clustering and momentum contrastive learning**

**Rating:** 7
**Confidence:** 3

**Review:**

[Overview]
In this paper, the authors tackle the problem of unsupervised visual representation learning by combining the deep clustering technique and momentum contrastive learning. Contrastive learning is widely used in instance-level SSL. However, considering it cannot model the inter-sample structures, the authors proposed a new method called prototypical contrastive learning (PCL), which boosts the original contrastive learning with clustering technique. It can be interpreted as an EM procedure. Though the high-level idea is similar to deep clustering and its variants, the authors argued that PCL can address the issue of reinitialization and potential cluster collapse, as demonstrated in the formulas. In the experiments, the authors performed extensive experiments to demonstrate the efficiency of PCL.

[Strength]
1. The authors presented a well-written paper with some theoretical explanations and extensive experiments. It is easy to follow with some good insights in the words.
2. The proposed PCL is formulated as an EM procedure. Though see it as an EM algorithm at a high-level is not novel, this paper has a relatively thorough derivation and give a good intuition behind the proposed PCL method.
3. The authors demonstrated the effectiveness of the proposed PCL on various datasets and settings. Compared with MoCo, its main counterpart, PCL achieve substantial improvement over it across the board.

[Concerns]
1. Introducing a EM-like procedure to mitigate the reinitialization issue has been proposed in previous works. Though not on large-scale dataset, JULE (Yang et al 2016) proposed to use a forward and backward pass for alternative learning, which corresponds to the E-step and M-step in this paper, respectively. To date, there are also some papers that proposed a similar strategy to address the issue of oscillation, such as "Online Deep Clustering for Unsupervised Representation Learning. 2020".
2. The proposed PCL combined momentum contrast and deep cluster technique, as formulated in Eq. (11). This formula prompts the interaction of intra-instance learning and inter-instance learning. From the experiments, it is not clear how they cooperate with each other during the training. Does the PCL (2nd term in Eq. (11)) solely perform poorly in representation learning?

[Summary]
Overall, I think this paper presents a clean formula to reconcile the intra-instance and inter-instance unsupervised learning methods, which seems to be a promising direction for further exploration. Based on my above comments, I think this is a good paper, and recommend acceptance. I would suggest the authors can address my above concerns and provide more insights about how these two losses interact with each other during the learning process in the rebuttal.

---

> ### Author Response · Authors · 2020-11-14
> **Response to Reviewer #4**
>
> We thank the reviewer for the insightful and positive comments. Here are our responses.
>
> We agree with the reviewer that joint clustering and representation learning has been proposed in the previous methods. Here we want to emphasize that the contribution of PCL is to bridge clustering with contrastive learning, and explicitly formulate the EM procedures. We appreciate the relevant online clustering paper and have discussed it in related works.
>
> We observe that both terms in Eq. (11) are important to the performance. The inter-instance learning (first term) is more important at the beginning of training, while the intra-instance learning (second term) becomes more important as training progresses and the clusters become more meaningful. In Table 7 (appendix A), we report the experimental results with only the first or the second term. We observe that using only the second term achieves decent results, given that the first term is used for warm-up. We hypothesize that a better training curriculum could further improve the performance of PCL, but leave it for future work.

---

### Decision · Program_Chairs · 2021-01-07
**Final Decision**

**Decision:**

Accept (Poster)

**Comment:**

This paper proposes an extension to previous unsupervised feature learning work, with an EM-style latent variable model with momentum encoders. The paper is well-written and provides a nice read. It has been noted that it is easy to follow and provides good insights. On the experimental side, compared with MoCo, the proposed approach achieve noticeable improvements. One of the reviewers noted the easy reproducibility of the proposed approach.

Some reviewers noted some comparisons were lacking from the original manuscript, but the authors have update the draft to include those. As noted in the reviews, the field of SSL in vision is moving at a very quick pace, making it hard to clearly state what is the SOTA at time t.

Overall, most questions raised by the reviewers were properly addressed during rebuttal - and given the ratings, I suggest acceptance.